# Mediating Factors Explaining the Associations between Solid Fuel Use and Self-Rated Health among Chinese Adults 65 Years and Older: A Structural Equation Modeling Approach

**DOI:** 10.3390/ijerph19116904

**Published:** 2022-06-05

**Authors:** Qiutong Yu, Yuqing Cheng, Wei Li, Genyong Zuo

**Affiliations:** 1Centre for Health Management and Policy Research, School of Public Health, Cheeloo College of Medicine Shandong University, 44 Wen-Hua-Xi Road, Jinan 250012, China; yuqiutong@126.com (Q.Y.); yuqing686@163.com (Y.C.); liwei0830zjqz@163.com (W.L.); 2NHC Key Laboratory of Health Economics and Policy Research, Shandong University, 44 Wen-Hua-Xi Road, Jinan 250012, China

**Keywords:** household air pollution, solid fuel, self-rated health, structural equation modeling, elderly

## Abstract

Exposure to indoor air pollution from cooking with solid fuel has been linked with the health of elderly people, although the pathway to their association is unclear. This study aimed to investigate the mediating effects between solid fuel use and self-rated health by using structural equation modeling (SEM) with the baseline data from Chinese Longitudinal Healthy Longevity Survey (CLHLS). We conducted a cross-sectional survey among 7831 elderly people aged >65 years from the CLHLS. SEM was used to analyze the pathways underlying solid fuel use and self-rated health. We estimated indirect effects of sleep quality (β = −0.027, SE = 0.006), cognitive abilities (β = −0.006, SE = 0.002), depressive symptoms (β = −0.066, SE = 0.007), systolic blood pressure (β = 0.000, SE = 0.000), and BMI (β = −0.000, SE = 0.000) on the association between solid fuel and the self-rated health using path analysis. Depressive symptoms emerged as the strongest mediator in the relationship between solid fuel use and self-rated health in the elderly. Interventions targeting sleep quality, cognitive abilities, depressive symptoms, systolic blood pressure, and BMI could greatly reduce the negative effects of solid fuel use on the health of the elderly population.

## 1. Introduction

The aging population in China has grown rapidly in the past few decades. According to the report of the seventh census in 2020, the population of those aged >60 years was 264.02 million, accounting for 18.70% of the total population, while the population of those aged >65 years was 190.64 million, accounting for 13.50% [1]. The Chinese population is approaching the depth of the aging stage. With the deepening of social aging in China, more attention should be paid to both physical and mental health problems in elderly people. 

Approximately 490 million people in China are exposed to indoor air pollution from cooking with solid fuel, such as coal, charcoal, and wood [2]. The particulate matters produced by burning solid fuel, such as PM2.5, PM10, carbon monoxide, nitrogen dioxide, sulfur dioxide, or other volatile organic compounds [3], have a negative impact on the physical or mental health of elderly people [4,5,6]. Therefore, it is necessary to explore the effect of indoor air pollution caused by using solid fuel on the health of elderly people.

Previous studies have shown that the use of solid fuel seriously affected both the mental and the physical health of the elderly [5,7,8,9,10,11]. Many studies on middle-aged and elderly people have concluded that solid fuel use was significantly correlated with the health of elderly people, in terms of poor sleep quality [7], low cognitive function [8], high incidence of arthritis [12], high incidence of depression [5], and high incidence of hypertension [9,10]. In pathogenesis research, there is an increasing link between indoor air pollution and physical diseases from solid fuel [13]. The underlying mechanism is that solid fuel in the combustion chamber produces toxic volatile organic compounds (VOCs), which can easily turn into vapors and are involved in metabolic processes that lead to low cognitive function or an increased blood pressure [8,14,15]. However, all the abovementioned effects are single indicators of health measurement. Self-rated health is a comprehensive measurement indicator of health that can reflect the respondents’ physiological state, their knowledge of this state, and their health expectations [16,17]. However, the association of solid fuel use with self-rated health in elderly people is unclear. Moreover, evidence on the effect of solid fuel use on the self-rated health of elderly people directly and indirectly through multiple mediators and the distinctive pathways, particularly in China, is lacking.

The available evidence suggests that people with long-term exposure to indoor air pollution from cooking with solid fuel are more likely to have poor sleep quality [7], high risk of depressive symptoms [5], low cognitive function [8], high blood pressure [9,10], and low body mass index (BMI) [11], which might result in poor self-rated health as mediators in the elderly. Therefore, structural equation modeling (SEM) was used to evaluate the total, direct, and indirect effects of exposure to indoor air pollution from cooking with solid fuel on the self-rated health in a mediation analysis and to assess the indirect effect within these distinctive paths.

This study aimed to investigate the mediating effects between solid fuel use and self-rated health using SEM with the baseline data from the Chinese Longitudinal Healthy Longevity Survey (CLHLS). Changing the fuel for cooking is a long-term project and may be costly, which may impose a huge financial burden on developing countries such as China. However, we can change the mediating factors to minimize the danger of solid fuel use to the health of the elderly, which could be of great help to achieve healthy aging in countries that are evolving into an aging society. 

## 2. Materials and Methods

### 2.1. Setting and Participants

We used secondary data derived from the 2018 CLHLS, which has been a cohort project since 1998, to conduct a longitudinal population-based study of people aged >65 years in China. We used the 2018 CLHLS because it contained the latest data which best fit the current situation of China’s aging society. Using the multistage stratified proportional probability sampling design, approximately 16,000 elderly people in urban and rural communities were randomly selected from 500 sample areas in 23 provinces, and 15,874 people were interviewed successfully. The biomedical ethics committee of Peking University approved the study, and all study participants signed an informed consent form. After excluding 95 participants who were younger than 65 years, 45 participants who “never cook”, 51 participants who had technical problems, 2409 participants who refused to answer, 1103 participants who answered “not applicable”, and 4340 participants who were unable to provide the data, 7831 individuals were finally included in the study (Figure 1).

### 2.2. Outcome Variables

Respondents were asked, “how do you feel about your health?”, and, according to the five-point Likert scale, their health status was rated as “very good”, “good”, “fair”, “poor”, or “very poor”. Self-rated health status has been identified as a reliable predictor of health and has been widely used in previous health studies conducted in China [18].

### 2.3. Exposure Variables

Respondents were asked, “what is the main source of cooking fuel in your family?”, and those who answered “other” were excluded. We defined coal, charcoal, and wood as solid fuel, whereas solar energy, natural gas, induction cooker, and other electrical appliances were defined as clean fuel. 

### 2.4. Mediators

Previous studies [5,7,8,9,10,11] have found that solid fuel use affects sleep quality, cognition abilities, depressive symptoms, blood pressure, and BMI. Respondents were asked “how is your sleep quality?” according to a five-point Likert scale, and sleep quality was rated as “very good”, “good”, “moderate”, “poor”, or “very poor”. Blood pressure was the average of two measurements on the right arm of participants using a mercury sphygmomanometer after the participants had rested for 5 min. Body mass index was calculated as the weight in kilograms divided by the height in meters squared (kg/m^2^).

Cognitive function was measured using the Chinese version of MMSE (the Mini Mental State Examination) [19]. The MMSE has been validated in previous studies for the Chinese elderly [20,21]. The correct answers were encoded as 1, while incorrect answers or “inapplicable” answers were encoded as 0. Then, we summed the cognitive scores of each participant. Higher cognitive scores indicated better cognitive function.

Depression was measured using the questions from the 10-item Center for Epidemiologic Studies Short Depression Scale (CES-D scale) [22], which has been translated and validated in previous studies for assessing cognitive levels in the Chinese elderly [23]. There are five levels of answers to all questions; we reverse-coded the positively oriented questions and recoded all responses as follows: “always” as 5, “often” as 4, “sometimes” as 3, “seldom” as 2, and “never” as 1. The depression scores of the participants were summed. Higher depression scores indicated greater depression severity.

### 2.5. Covariates

Notably, many studies have estimated the modification effects by sociodemographic factors and lifestyle behaviors when exploring the effects of household air pollution on the mediators in this paper [24,25]. Demographic characteristics including age (years), sex (female/male), and marital status (not married and married) were analyzed. The “not married” status included widowed, divorced, separated, or never married. Community location (urban/rural) and socioeconomic status, including education and family income, were also analyzed. Education was recoded into three levels (0 years/1–6 years/<6 years). Family income was recoded into three levels (>10,000 CNY, 10,000–50,000 CNY, and <50,000 CNY). Lifestyle behaviors including smoking status (not current/current), alcohol use (not current/current), and regular exercise (not current/current) were also evaluated [26].

### 2.6. Statistical Analysis

The chi-square test was used for dichotomous variables, including sex, marital status, community location, education years, household income, smoking status, alcohol use, physical activity, sleep quality, and self-rated health, and the *t*-test was used for continuous variables, including cognitive abilities, depressive symptoms, systolic blood pressure, and BMI between those who cook with solid fuel and those who cook with clean fuel.

Structural equation modeling (SEM) is a multivariate analytic technique. It is used to simultaneously assess multiple relationships among variables. SEM was used to conduct a formal mediation test and disaggregate the relationship between solid fuel use and self-rated health through causally defined indirect and direct pathways. SEM contains a series of multiple regression models, linear regression models for continuous outcomes, and logistic regression models for binary outcomes. The proportion of the total effect of solid fuel on self-rated health attributable to the mediators was calculated by dividing the ratio of the indirect effect through the mediated pathway by the ratio of the total effect. Estimation for SEM was performed using maximum likelihood. Three common measures were used to evaluate the fit indices of SEM: the root-mean-square error of approximation (RMSEA), comparative fit index (CFI), and Tucker–Lewis index (TLI). TLI and CFI values of 0.95 indicate a reasonably good fit [27]. An RMSEA value of 0.05–0.08 represents a moderate fit, while a value of 0.08–0.10 represents an acceptable fit [28]. All data were analyzed using Stata15.0.

## 3. Results

### 3.1. Basic Characteristics of the Participants

The characteristics of the study participants are shown in Table 1. Of the 7831 individuals, 2317 (29.59%) used solid fuel for cooking, and 5514 (70.41%) used clean fuel for cooking. Significant differences in sex, marital status, community location, education years, income, smoking status, alcohol use, physical activity, sleep quality, cognitive abilities, depressive symptoms, hypertension, BMI, and self-rated health were observed between individuals using solid fuel and those using clean fuel for cooking (*p* < 0.05). Participants who used clean fuel had higher socioeconomic indicators, in terms of both years of education and family income, than those using indoor solid fuel. Moreover, the proportions of individuals with current smoking (19.25% vs. 16.00%) and drinking (17.22% vs. 15.43%) habits were higher among individuals using solid fuel for cooking than among those using clean fuel. However, the proportion of individuals who engaged in physical activities was significantly higher among those using clean fuel for cooking (42.00% vs. 24.08%) than among those using solid fuel. Moreover, individuals using clean fuel for cooking reported better sleep quality, cognitive abilities, and self-rated health. Among those using solid fuel for cooking, the average systolic blood pressure was 141.07 mmHg.

### 3.2. Structural Equation Model

Structural equation modeling is a multivariate analytic technique used to simultaneously assess multiple relationships among variables. As shown in Figure 2, we performed SEM with a good fit to the data (RMSEA = 0.045, CFI = 0.970, TLI = 0.829), showing that the model fit quite well after adjusting for age, sex, marital status, community location, education years, household income, smoking status, alcohol use, and physical activity. To determine the extent of the impact of solid fuel, sleep quality, cognitive abilities, depressive symptoms, systolic blood pressure, and BMI on the self-rated health of the elderly people in the path model, a standardized path coefficient of the SEM was estimated (Table 2). The total indirect effect of solid fuel use for self-rated health was −0.146 (*p* < 0.001). A significant direct effect of the use of solid fuel on self-rated health (β = −0.041, S.E. = 0.020), with indirect effects accounting for most of the total effects, was identified.

Table 2 shows that sleep quality (β = 0.250), cognitive ability (β = −0.010), and depression (β = −0.049) had a direct influence on self-rated health (*p* < 0.005). The result from SEM indicated that using solid fuel exhibited a direct effect on sleep quality (β = −0.056), cognitive ability (β = −0.319), depression (β=0.910), and BMI (β = −0.733). However, we did not find that systolic blood pressure (*p* = 0.876) and BMI (*p* = 0.876) were significantly associated with self-rated health, nor did we find that solid fuel was significantly associated with systolic blood pressure. Moreover, SEM is used either to assess the total effect (i.e., direct and indirect effects) of a treatment or exposure on an outcome in the mediation analysis or to assess a specific indirect effect with those complex paths. First, a significant negative indirect effect of solid fuel use on self-rated health via sleep quality was observed (= −0.013, SE = 0.006). Second, cognitive abilities were also found to be a mediator between solid fuel use and self-rated health (β = −0.003, SE = 0.001). Third, a significant negative indirect effect of solid fuel on self-rated health via depressive symptoms (β = −0.045, SE = 0.007) was also observed. Additionally, the indirect effects of systolic blood pressure on self-rated health, as well as those of body mass index on self-rated health, were not detected in the model involving solid fuel exposure and self-rated health (all *p* > 0.05). 

## 4. Discussion

### 4.1. Main Findings

In this cross-sectional study, we focused on assessing the potential mediating factors for the relationships between solid fuel use and self-rated health. Exposure to solid fuel was found to have a direct contribution to the decreased score of self-rated health. Moreover, we observed that exposure to solid fuel was significantly associated with a decreased score of self-rated health, and this linkage was mediated by sleep quality, cognitive ability, and depression symptoms. These effects remained significant even after controlling for confounders such as sociodemographic factors and lifestyle behaviors.

### 4.2. Available Evidence on the Association of Solid Fuel with Self-Rated Health

We found that the direct effect of solid fuel use on self-rated health was −0.044 (*p* < 0.001). Previous studies have demonstrated that solid fuels can affect physical health measures such as blood pressure, BMI, cognition, and mental health measures such as sleep quality and depression status. However, the dependent variables used in these studies were all single health indicators evaluating a specific aspect of the health of an individual. Nevertheless, by constructing a structural equation model, we found that solid fuel can affect the above single health indicators, thereby affecting the comprehensive indicators of personal health, because self-rated health can reflect people’s physical and mental health. Therefore, screening and management of these disorders in older adults with heavy use of solid fuels is necessary.

### 4.3. Depression Was the Strongest Mediator of the Relationship between Solid Fuel Use and Self-Rated Health

The results showed that individuals using solid fuel had greater depression severity (β = 0.893), and depression symptoms emerged as the strongest mediator of the relationship between solid fuel and self-rated health rather than sleep quality and cognitive ability. However, we only found three similar studies exploring the association between solid fuel and depression whose findings were consistent with our results that individuals using solid fuel were at a higher risk of depression [5,29,30]. Moreover, regarding household air pollution, there is an increasing link between mental diseases and household air pollutants. This study contributes to the limited literature on the association between solid fuel and poor sleep quality in the elderly. Our results indicated that using solid fuel was significantly associated with poor sleep quality, even after accounting for a wide range of covariates, including sociodemographic factors, lifestyle behaviors, and presence of chronic diseases. Our findings were in line with previous findings [31,32]. However, limited evidence is available on the association between mental diseases and household air pollutants in pathogenesis research. One possible reason is that solid fuel combustion produces much higher levels of various gaseous pollutants than clean fuel and may increase the risk of developing mental disorders such as depression and poor sleep quality through pathways such as cerebrovascular damage, oxidative stress, neuroinflammation, or neurodegeneration [33,34].

### 4.4. Cognition as the Protective Factor Linking Systolic Blood Pressure to Self-Rated Health 

An indirect path linking solid fuel and self-rated health was identified through cognitive abilities, although the other mediators including systolic blood pressure and BMI were not found to be associated with self-rated health. Our findings suggested that people who use solid fuel for cooking had lower cognitive abilities (β = −0.319, SE = 0.111). Several studies in China contributed to the literature on the association between solid fuel and poor cognitive ability in the elderly. Recent evidence from a study in China found that solid fuel use was significantly associated with mental health and cognitive ability in middle-aged and older adults [8]. There were also studies that further showed the potentially harmful effects of household air pollution exposure on other aspect of memory. A follow-up study showed that solid fuel use was associated with a greater decline in cognitive score, mostly in the episodic memory and visuo-construction dimensions [35]. Another prospective analysis found a significant adverse impact on cognitive abilities, especially short-term memory and mathematical reasoning. These results demonstrated that using solid fuel poses a health threat to elderly people [36].

### 4.5. Strengths and Limitations

This study had several strengths. Firstly, this study specified the pathways of the relationship between solid fuel use and self-rated health of the elderly. Secondly, self-rated health was not only considered a comprehensive measurement indicator of health, but also used as a predictor of morbidity and mortality [37]. This study’s results can help achieve primary and secondary prevention and improve the health of the elderly by stopping the use of solid fuel. Lastly, our study included 500 sample areas in 23 provinces in China, which gives our findings strong external validity for the Chinese society. Moreover, the findings were robust across the study regions, demographic characteristics, and lifestyle behaviors.

However, a major drawback should be noted. The assessment of self-rated health in our study was based on only one self-reported question, and this may not provide exactly accurate information compared with a structured interview tool. Other limitations should also be mentioned. Firstly, constrained by the CLHLS data, we were unable to derive information on whether the individuals were responsible for cooking, types of cooking in childhood, and time spent cooking with indoor solid fuel; additionally, indoor air pollution exposure may vary by family and personal characteristics [38]. Secondly, the results of this cross-sectional study may not explain the underlying mechanisms of the relationship between solid fuel use and self-rated health. The underlying mechanisms may need to be investigated in large prospective cohort studies. Thirdly, the relationship of mediators including sleep quality, cognitive ability, depression, systolic blood pressure, and BMI with solid fuel use has been confirmed in previous studies. Other diseases or pathophysiological indicators that may be caused by the use of solid fuel, which have not been proven, can be explored in future studies. 

## 5. Conclusions

Sleep quality, cognitive abilities, and depressive symptoms partially contributed to the association between solid fuel use and self-rated health. However, systolic blood pressure and BMI were not found to be directly associated with self-rated health. Among these mediators, depression was the strongest mediator of the relationship between solid fuel use and self-rated health. Our results demonstrated that using solid fuel poses a health threat for elderly people. Replacing solid fuel with clean fuel may be an important way to improve self-rated health of elderly people. Regarding this, priority should be giving to those with significant solid fuel exposure.

## Figures and Tables

**Figure 1 ijerph-19-06904-f001:**
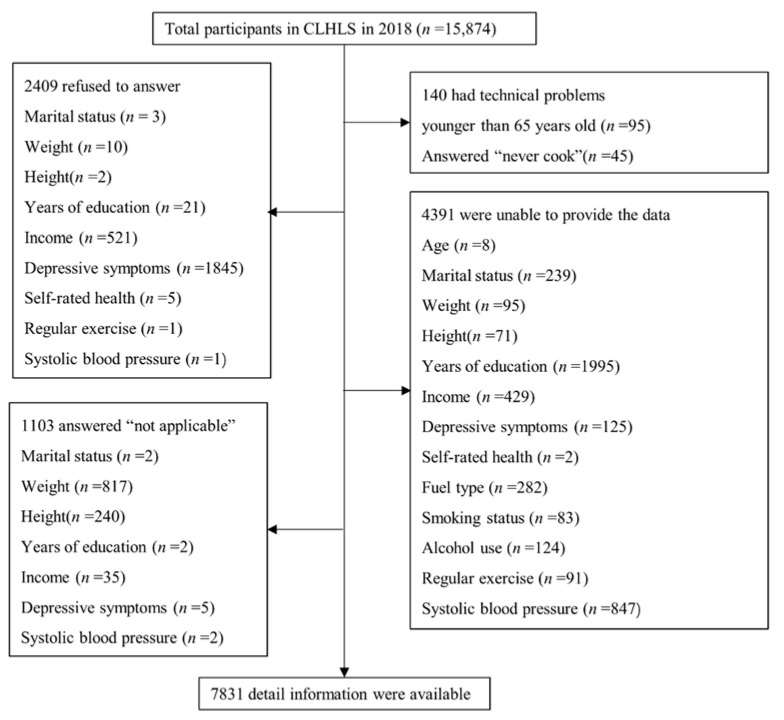
Study flowchart of participant selection (aged ≥65 years) from the Chinese Longitudinal Healthy Longevity Survey 2018 survey data. Abbreviation: CLHLS: Chinese Longitudinal Healthy Longevity Survey.

**Figure 2 ijerph-19-06904-f002:**
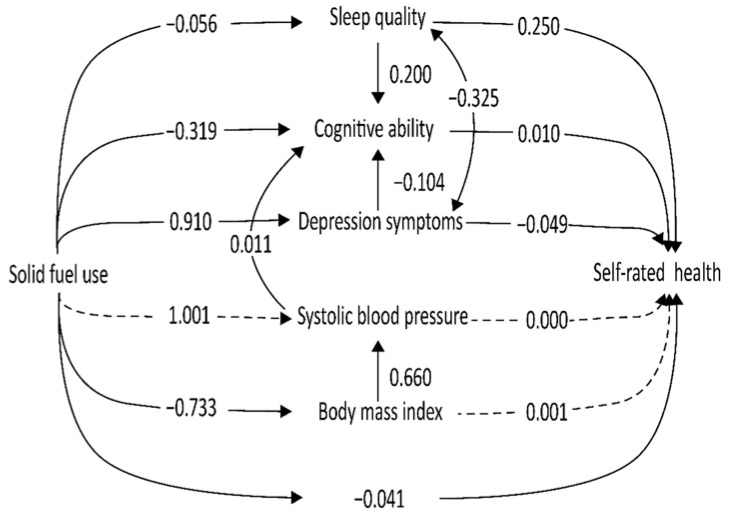
Pathways between solid fuel, mediators, and self-rated health, according to the Chinese Longitudinal Healthy Longevity Survey, 2018. Structural equation modeling was performed among Chinese older adults over 65 years old. The model was adjusted for age, sex, marital status, community location, education years, household income, smoking status, alcohol use, and physical exercise. Dashed lines denote insignificant pathways between solid fuel, mediators, and self-rated health, while solid lines denote significant pathways between solid fuel, mediators, and self-rated health.

**Table 1 ijerph-19-06904-t001:** Characteristics of selected variables among the participants.

Variables	All (*n* = 7831)	Solid Fuel (*n* = 2317)	Clean Fuel (*n* = 5514)	*p*-Value
Age (years, mean, SD)	82.70 (11.28)	82.75 (11.22)	82.68 (11.31)	0.783
Sex (*n*, %)				0.024
Male	3682 (47.02)	1044 (45.06)	2638 (47.84)	
Female	4149 (52.98)	1273 (54.94)	2876 (52.16)	
Marital status (*n*, %)				0.024
Not married	3997 (51.04)	1137 (49.07)	2860 (51.87)	
Married	3834 (48.96)	1180 (50.93)	2654 (48.13)	
Community Location (*n*, %)				<0.001
Urban	4383 (55.97)	907 (39.15)	3476 (63.04)	
Rural	3448 (44.03)	1410 (60.85)	2038 (36.96)	
Education (*n*, %)	3407 (43.51)	1272 (54.90)	2135 (38.72)	<0.001
0	2451 (31.30)	421 (18.17)	2030 (36.82)	
1–6	1973 (25.19)	624 (26.93)	1349 (24.46)	
7–				<0.001
Income (n, %)	2105 (26.88)	1055 (45.53)	1050 (19.04)	
0–10,000	2633 (33.62)	884 (38.15)	1749 (31.72)	
10,000–50,000	3093 (39.05)	378 (16.31)	2715 (49.24)	
≥50,000				
Smoking status (*n*, %)				<0.001
Not current	6503 (83.04)	1871 (80.75)	4632 (84.00)	
Current	1328 (16.96)	446 (19.25)	882 (16.00)	
Alcohol use (*n*, %)				0.049
Not current	6581 (84.04)	1918 (82.78)	4663 (84.57)	
Current	1250 (15.96)	399 (17.22)	851 (15.43)	
Regular exercise (*n*, %)				<0.001
Not current	4957 (63.30)	1759 (75.92)	3198 (58.00)	
Current	2874 (36.70)	558 (24.08)	2316 (42.00)	
Sleep quality (*n*, %)				<0.001
Very poor	160 (2.04)	47 (2.03)	113 (2.05)	
Poor	926 (11.82)	301 (12.99)	625 (11.33)	
Moderate	2471 (31.55)	801 (34.57)	1670 (30.29)	
Good	2939 (37.53)	854 (36.86)	2085 (37.81)	
Very good	1335 (17.05)	314 (13.55)	1021 (18.52)	
Cognitive abilities (mean, SD)	20.44 (4.93)	19.90 (5.29)	20.67 (4.75)	<0.001
Depression symptoms (mean, SD)	22.87 (5.38)	24.05 (5.44)	22.37 (5.28)	<0.001
SBP ^1^ (mmHg, mean, SD)	139.50 (20.82)	141.07 (21.56)	138.84 (20.46)	<0.001
BMI (kg/m^2^, mean, SD)	22.67 (4.36)	22.02 (3.92)	22.94 (4.50)	<0.001
Self-rated health (*n*, %)				<0.001
Very poor	63 (0.80)	27 (1.17)	36 (0.65)	
Poor	857 (10.94)	299 (12.90)	558 (10.12)	
Fair	2969 (37.91)	949 (40.96)	2020 (36.63)	
Good	2892 (36.93)	813 (35.09)	2079 (37.70)	
Very good	1050 (13.41)	229 (9.88)	821 (14.89)	

^1^ SBP, systolic blood pressure; BMI, body mass index; SD, standard deviation.

**Table 2 ijerph-19-06904-t002:** Direct, indirect, and total effects of solid fuel use on self-rated health.

Pathways	β ^1^	SE	*p*-Value
**Direct effects**			
Solid fuel→sleep quality	−0.056	0.026	0.030
Solid fuel→cognitive abilities	−0.319	0.111	0.004
Solid fuel→depression symptoms	0.910	0.138	<0.001
Solid fuel→ systolic blood pressure	1.001	0.556	0.068
Solid fuel→body mass index	−0.733	0.111	<0.001
Solid fuel→self-rated health	−0.041	0.020	0.037
Sleep quality→self-rated health	0.250	0.009	<0.001
Cognitive abilities→self-rated health	0.010	0.002	<0.001
Depression symptoms→self-rated health	−0.049	0.002	<0.001
Systolic blood pressure→self-rated health	0.000	0.000	0.876
Body mass index→self-rated health	0.001	0.002	0.599
**Indirect effects**			
Solid fuel→sleep quality→self-rated health	−0.013	0.006	0.030
Solid fuel→cognitive symptoms→self-rated health	−0.003	0.001	0.011
Solid fuel→depression symptoms→self-rated health	−0.045	0.007	<0.001
Solid fuel→systolic blood pressure→self-rated health	0.000	0.000	0.0877
Solid fuel→body mass index→self-rated health	−0.001	0.002	0.600
**Total effects**			
Solid fuel→self-rated health	−0.107	0.022	<0.001

^1^ β, coefficient; SE, standard error.

## Data Availability

Data are available on the open research data service platform of Peking University. Data for this study were sourced from the Chinese Longitudinal Healthy Longevity Survey (CLHLS) and are available at https://opendata.pku.edu.cn (accessed on 1 January 2022).

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
