# Peer review of "Mediating Factors Explaining the Associations between Solid Fuel Use and Self-Rated Health among Chinese Adults 65 Years and Older: A Structural Equation Modeling Approach"

_ijerph, 2022, doi:10.3390/ijerph19116904_

Round 1

Reviewer 1 Report

Elaborate a little more on SEM, and try to explain with clarity in result section. 

Author Response

  1. Elaborate a little more on SEM, and try to explain with clarity in result section. 

[Response]: Thank you for your valuable suggestion.

First, we elaborated Structural equation model in the Statistical analysis section, “Structural equation modeling (SEM) is a multivariate analytic technique. It is used to assess simultaneously multiple relationships among variables. SEM was used to conducted a formal mediation test and disaggregate the relationship between solid fuel use and self-rated health through causally defined indirect and direct pathways. SEM contains a series of multiple regression models, linear regression models for continuous outcomes, and logistic regression models for binary outcomes. The proportion of the total effect of solid fuel on self-rated health that is attributable to the mediators was calculated by dividing the ratio of the indirect effect through the mediated pathway by the ratio of the total effect. Estimation for SEM was performed using maximum likelihood. Three common measures were used to evaluate the fit indices of SEM: the root mean square error of approximation (RMSEA), comparative fit index (CFI), and Tucker–Lewis index (TLI). The TLI and CFI value of 0.95 indicates a reasonably good fit [1]. The RMSEA value of 0.05–0.08 represents a moderate fit, and 0.08–0.10 represents an acceptable fit [2]. All data were analyzed using Stata15.0.

Second, we revised the result section as follows.

3.1 Basic characteristics of the participants

The characteristics of the study participants are shown in Table 1. Of the 7831 individuals, 2317 (29.59%) used solid fuel for cooking, and 5514 (70.41%) used clean fuel for cooking. Significant differences in sex, marital status, community location, education years, income, smoking status, alcohol use, physical activity, sleep quality, cognitive abilities, depressive symptoms, hypertension, BMI, and self-rated health were observed between individuals using solid fuel and those using clean fuel for cooking (p<0.05). Participants who use clean fuel had higher socioeconomic indicators, both in terms of years of education and family income, than those using indoor solid fuel. Moreover, the proportions of individuals with current smoking (19.25% vs. 16.00%) and drinking (17.22% vs. 15.43%) habits were higher among individuals using solid fuel for cooking than among those using clean fuel. However, the proportion of individuals who engaged in physical activities was significantly higher among those using clean fuel for cooking (42.00% vs. 24.08%) than among those using solid fuel. Moreover, individuals using clean fuel for cooking reported better sleep quality, cognitive abilities, and self-rated health. Among those using solid fuel for cooking, the average systolic blood pressure reached 141.07 mmHg.

3.2. Structural equation model

Structural equation modeling is a multivariate analytic technique used to assess simultaneously multiple relationships among variables. As shown in Fig. 2, we performed SEM with a good fit to the data (RMSEA=0.045, CFI=0.970, TLI=0.829), which shows that the model fits quite well, after adjusting for age, sex, marital status, community location, education years, household income, smoking status, alcohol use, and physical activity. To determine the extent of the impact of solid fuel, sleep quality, cognitive abilities, depressive symptoms, systolic blood pressure, and BMI on the self-rated health of the elderly people in the path model, a standardized path coefficient of the SEM was estimated (Table 2). The total indirect effect of solid fuel use for self-rated health was -0.146 (P<0.001). A significant direct effect of the use of solid fuel on self-rated health (b=-0.041, S.E.=0.020), with indirect effects accounting for most of the total effects, was identified.

Table 2 shows that sleep quality (b=0.250), cognitive ability (b=-0.010), and depression (b=-0.049) had a direct influence on self-rated health (P<0.005). The result from SEM indicated that using solid fuel exhibited a direct effect on sleep quality (b=-0.056), cognitive ability (b=-0.319), depression (b=0.910), and BMI (b=-0.733). However, we did not find that systolic blood pressure (P=0.876) and BMI (P=0.876) were significantly associated with self-rated health, nor did we find that solid fuel was significant associated with systolic blood pressure. Moreover, SEM is used either to assess the total effect (i.e., direct and indirect effects) of a treatment or exposure on an outcome in the mediation analysis or to assess specific indirect effect with those complex paths. First, a significant negative indirect effect of solid fuel use on self-rated health via sleep quality was observed (b=−0.013, S.E.=0.006). Second, cognitive ability was also found to be a mediator between solid fuel use and self-rated health (b=−0.003, S.E.=0.001). Third, a significant negative indirect effect of solid fuel on self-rated health via depressive symptoms (b=−0.045, S.E.=0.007) was also observed. Additionally, the indirect effects of systolic blood pressure on self-rated health as well as that of body mass index on self-rated health were not detected in the model involving solid fuel exposure and self-rated health (all p>0.05).

Reviewer 2 Report

This paper examines the association between solid fuel use and self-reported health in elderly (age >65) Chinese adults. The subject matter is of considerable public health interest and this study has the merit of being based on a large and fairly representative sample. However, the following points require clarification or correction (as appropriate) on the part of the authors:

  1. It is not clear what is meant by "4340 participants missed the data" (p 2, line 79). Does this mean that these participants were unable to provide information, or that it was provided but could not be retrieved later? This sentence could be rephrased to make the intended meaning clearer.
  2. In the introduction, the authors report correlations between solid fuel use and several health outcomes (depression, arthritis, hypertension, etc.) What are the possible causal mechanisms for these associations? Are there any key confounding factors that require consideration when examining such associations? This should be discussed in the Introduction and Discussion sections as appropriate.
  3. Figure 1 should be provided in the standard format for observational studies. All abbreviations used in the figure should be explained in a footnote.
  4. The use of a single self-rated measure of health has both merits and demerits. What other methods did the researchers consider when selecting the key outcome measure for their study? How else could global health have best been captured? Was any attempt made to validate this measure (e.g. with random measurements of blood pressure or symptom self-ratings followed by correlation with self-report "global" scores)? If not, this should be discussed as a significant limitation of the current study.
  5. Details of translation / validation of the translated instrument should be provided for the MMSE and CES-D. If this has already been done by earlier researchers, these papers should be cited in the text.
  6. The statistical model selected by the authors is questionable. The "mediating factors" they have included in their analysis are health outcomes themselves, and have been associated with fuel use / indoor air pollution by earlier researchers. A more accurate mediation analysis would include the effects of truly independent factors (e.g. age,  sex, socioeconomic status, genotype, tobacco use health conditions / diseases not known to be associated with solid fuel use) as mediating factors. The use of factors themselves related to fuel use (depression, cognition, sleep, blood pressure) as "mediators" compromises the analysis and limits the validity of any final conclusions drawn.
  7.  As many of the "mediators" are themselves domains of health, it would be expected that they would correlate significantly with a rating of self-reported overall health. Thus, it is not clear what is being proved or demonstrated even if the study findings are statistically significant, beyond a partial replication of earlier results. Of course, replications have significant value as research findings per se, but it is unclear what else the current study was attempting to demonstrate.
  8. The following statement illustrates the problems discussed in #6 and #7 above: "Replacing solid fuel in cooking in a population of 490 million people would be a huge financial undertaking for China, and improving the mental health of the elderly could significantly improve their self-rated health." (p 8, lines 229-231). If depression is itself related to indoor pollution / solid fuel use, then treating it without modifying the pertinent environmental risk factor(s) would have limited success. In other words (as mentioned in #6 above) if solid fuel use and depression are not independent of each other, it is unlikely that "treating depression" would reduce the impacts of solid fuel use on health. This conclusion arises because of the flaws in the selection of mediating variables in the analysis and may not have real-world relevance.
  9. Limitations of the study should be discussed in more depth in the light of the points raised above (#4, #6, #7)
  10. A better conclusion would be to acknowledge the multi-faceted impact of solid fuel use / indoor air pollution on health (mental health, cognition, cardiovascular health, etc) and to highlight the need for better screening and management of these disorders in those with significant solid fuel use / exposure. The conclusions provided go against the public health perspective and suggest that the effect of exposure to pollutants can be attenuated by sleep hygiene, psychiatric treatment, etc. - which is very unlikely to be the case.

Author Response

  1. It is not clear what is meant by "4340 participants missed the data" (p 2, line 79). Does this mean that these participants were unable to provide information, or that it was provided but could not be retrieved later? This sentence could be rephrased to make the intended meaning clearer.

[Response]: I apologize for this confusion. We revised the sentence to “4340 participants who were unable to provide the data.” in P 2, L 84-85.

  1. In the introduction, the authors report correlations between solid fuel use and several health outcomes (depression, arthritis, hypertension, etc.) What are the possible causal mechanisms for these associations? Are there any key confounding factors that require consideration when examining such associations? This should be discussed in the Introduction and Discussion sections as appropriate.

[Response]: Thank you for your comment. First, we added the possible causal mechanisms in paragraph 3 “In pathogenesis research, there is an increasing link between indoor air pollution and physical diseases from solid fuel [1]. The underlying mechanism is that solid fuel in the combustion chamber produces toxic volatile organic compounds (VOCs), which can easily turn into vapors and are involved in metabolic processes that lead to low cognitive function or an increased blood pressure [2-4].” (P2, L45-50)

Second, we did our best to explain why solid fuel leads to poor sleep quality and depression in the Discussion section, “However, limited evidence is available on the association between mental diseases and household air pollutants in pathogenesis research. One possible reason is that solid fuel combustion produces much higher levels of various gaseous pollutants than clean fuel and may increase the risk of developing mental disorders such as depression and poor sleep quality through pathways such as cerebrovascular damage, oxidative stress, neuroinflammation, or neurodegeneration[5, 6]. ” (P8, L240-246)

Third, according to your suggestion, key confounding factors mentioned in previous studies including socio-demographic factors (age, sex, marital status, community location, education years and household income) and lifestyle behaviors (smoking status, alcohol use and regular exercise) have been incorporated into the structural equation model [7-9] and this has been written in the covariates section (P4, L118-120)

  1. Figure 1 should be provided in the standard format for observational studies. All abbreviations used in the figure should be explained in a footnote.

[Response]: We apologize for this mistake. Figure 1 in the manuscript uses TIFF format and is uploaded separately in a single zip archive format. All abbreviations have been explained in a footnote.

  1. The use of a single self-rated measure of health has both merits and demerits. What other methods did the researchers consider when selecting the key outcome measure for their study? How else could global health have best been captured? Was any attempt made to validate this measure (e.g. with random measurements of blood pressure or symptom self-ratings followed by correlation with self-report "global" scores)? If not, this should be discussed as a significant limitation of the current study.

[Response]: Thank you for your comment. In this study, we used secondary data from the 2018 Chinese Longitudinal Healthy Longevity Survey, and self-rated health was assessed with the question "How do you feel about your current state of health?". We had no other methods to measure self-rated health; therefore, we added this as a limitation of the study in the Strength and limitation section, “However, a major drawback should be noted. The assessment of self-rated health in our study was based on only one self-reported question, and this may not provide an exactly accurate information compared with a structured interview tool, although many studies also used this method to measure self-rated health.” (P9, L272-275)

  1. Details of translation / validation of the translated instrument should be provided for the MMSE and CES-D. If this has already been done by earlier researchers, these papers should be cited in the text.

[Response]: Thank you for your valuable suggestion. We added the key point to the description of the mediator variables as follows: “Cognitive function was measured using the Chinese version of MMSE (the Mini Mental State Examination) [10]. MMSE has been validated in previous studies for Chinese elderly [11, 12].” (P4, L108-110) “Depression was measured using the questions from the 10-item Center for Epidemiologic Studies Short Depression Scale (CES-D scale) [13], and CES-D scale has been translated and validated in previous studies for assessing cognitive levels in Chinese elderly [14].” (P4, L113-115)

  1. The statistical model selected by the authors is questionable. The "mediating factors" they have included in their analysis are health outcomes themselves, and have been associated with fuel use / indoor air pollution by earlier researchers. A more accurate mediation analysis would include the effects of truly independent factors (e.g. age, sex, socioeconomic status, genotype, tobacco use health conditions / diseases not known to be associated with solid fuel use) as mediating factors. The use of factors themselves related to fuel use (depression, cognition, sleep, blood pressure) as "mediators" compromises the analysis and limits the validity of any final conclusions drawn.

[Response]: Thank you for your comment. The purpose of our research was to explore the ways through which solid fuel can ultimately affect self-rated health. Dependent variables in previous studies are single physical or mental health indicators. Self-rated health is a comprehensive indication of individual health [15, 16]. We explained this point in the strength and limitation section, “the relationship of mediators including sleep quality, cognitive ability, depression, systolic blood pressure, and BMI with solid fuel use has been confirmed in previous studies. Other diseases or pathophysiological indicators that may be caused by the use of solid fuel, which have not been proven, can be explored in future studies.” (P9, L282-286)

  1. As many of the "mediators" are themselves domains of health, it would be expected that they would correlate significantly with a rating of self-reported overall health. Thus, it is not clear what is being proved or demonstrated even if the study findings are statistically significant, beyond a partial replication of earlier results. Of course, replications have significant value as research findings perse, but it is unclear what else the current study was attempting to demonstrate.

[Response]: Thank you for your valuable suggestion. First, we used Pearson correlation analysis to verify the relationship between the dependent variable and the mediator. As shown in Table, the correlation coefficient between the dependent variables and the mediators is up to 0.355, which is within the acceptable range. Due to the large sample size, the P value was significant.

Table Correlation analysis of self-rated health and the mediators.

Variables

Self-rated health

Self-rated health

1.000

Sleep quality

0.323***

Depression symptoms

-0.355***

Cognitive ability

0.141***

Systolic blood pressure

-0.017

Body mass index

0.065***

***P<0.001, **P<0.01, *P<0.05.

Second, the significance of our research is that we explored the impact of solid fuel on self-rated health at the global health level and explore the specific impact pathways (mainly including sleep, cognition, and depression) through structural equation modeling. Most important, we validated the findings of previous studies, in which using solid fuel was shown to have an effect on the above-mentioned conditions.

  1. The following statement illustrates the problems discussed in #6 and #7 above: "Replacing solid fuel in cooking in a population of 490 million people would be a huge financial undertaking for China, and improving the mental health of the elderly could significantly improve their self-rated health." (p 8, lines 229-231). If depression is itself related to indoor pollution / solid fuel use, then treating it without modifying the pertinent environmental risk factor(s) would have limited success. In other words (as mentioned in #6 above) if solid fuel use and depression are not independent of each other, it is unlikely that "treating depression" would reduce the impacts of solid fuel use on health. This conclusion arises because of the flaws in the selection of mediating variables in the analysis and may not have real-world relevance.

[Response]: Thanks for your constructive suggestion. We realize that there is a problem in the paragraph, therefore we have revised the Depression was the strongest mediator of the relationship between solid fuel use and self-rated health section as follows: “The results showed that individuals using solid fuel had greater depression severity (b=0.893), and depression symptoms emerged as the strongest mediator of the relationship between solid fuel and self-rated health rather than sleep quality and cognitive ability. However, we have only found three similar studies exploring the association between solid fuel and depression whose findings were consistent with our results that individuals using solid fuel were at a higher risk of depression. [17-19] Besides, regarding household air pollution, there is an increasing link between mental diseases and household air pollutants. This study contributes to the limited literature on the association between solid fuel and poor sleep quality in the elderly. Our results indicated that using solid fuel was significantly associated with poor sleep quality, even after accounting for a wide range of covariates, including socio-demographic factors, lifestyle behaviors, and presence of chronic diseases. Our findings were in line with previous findings[20-22]. However, limited evidence is available on the association between mental diseases and household air pollutants in pathogenesis research. One possible reason is that solid fuel combustion produces much higher levels of various gaseous pollutants than clean fuel and may increase the risk of developing mental disorders such as depression and poor sleep quality through pathways such as cerebrovascular damage, oxidative stress, neuroinflammation, or neurodegeneration[5, 6].” (P8, L229-246)

  1. Limitations of the study should be discussed in more depth in the light of the points raised above

[Response]: Thank you for your comment. We have revised the Strength and limitation section as follows: “However, a major drawback should be noted. The assessment of self-rated health in our study was based on only one self-reported question, and this may not provide an exactly accurate information compared with a structured interview tool. Other limitations should also be mentioned. First, constrained by the CLHLS data, we were unable to derive information on whether the individuals were responsible for cooking, types of cooking in childhood, and time spent cooking with indoor solid fuel; additionally, indoor air pollution exposure may vary by family and personal characteristics [23]. Second, the results of this cross-sectional study may not explain the underlying mechanisms of the relationship between solid fuel use and self-rated health. The underlying mechanisms may need to be investigated in large prospective cohort studies. Third, the relationship of mediators including sleep quality, cognitive ability, depression, systolic blood pressure, and BMI with solid fuel use has been confirmed in previous studies. Other diseases or pathophysiological indicators that may be caused by the use of solid fuel, which have not been proven, can be explored in future studies.” (P8-9, L272-L286)

  1. A better conclusion would be to acknowledge the multi-faceted impact of solid fuel use / indoor air pollution on health (mental health, cognition, cardiovascular health, etc) and to highlight the need for better screening and management of these disorders in those with significant solid fuel use / exposure. The conclusions provided go against the public health perspective and suggest that the effect of exposure to pollutants can be attenuated by sleep hygiene, psychiatric treatment, etc. - which is very unlikely to be the case.

[Response]: Thank you for your important comment, which have helped us revise the Discussion section. Based on the above suggestions, we have revised it as follows:

4.1. Main findings

In this cross-sectional study, we focused on assessing the potential mediating factors for the relationships between solid fuel use and self-rated health. Exposure to solid fuel was found to have a direct contribution to the decreased score of self-rated health. Moreover, we observed that exposure to solid fuel was significantly associated with decreased score of self-rated health, and this linkage was mediated by sleep quality, cognitive ability, and depression symptoms. These effects remained significant even after controlling for confounders like socio-demographic factors and lifestyle behaviors. (P8, L208-215)

4.2. Available evidence on the association of solid fuel with self-rated health

We found that the direct effect of solid fuel use on self-rated health was -0.044 (P<0.001). Previous studies have demonstrated that solid fuels can affect physical health measures such as blood pressure, BMI, cognition, and mental health measures such as sleep quality and depression status. However, the dependent variables used in these studies are all single health indicators that evaluate a specific aspect of the health of an individual. Regardless, by constructing a structural equation model, we found that solid fuel can affect the above single health indicators, thereby affecting the comprehensive indicators of personal health, because self-rated health can reflect people's physical and mental health. Therefore, screening and management of these disorders in older adults with heavy use of solid fuels is necessary. (P7, L216-226)

4.3. Depression was the strongest mediator of the relationship between solid fuel use and self-rated health

The results showed that individuals using solid fuel had greater depression severity (b=0.893), and depression symptoms emerged as the strongest mediator of the relationship between solid fuel and self-rated health rather than sleep quality and cognitive ability. However, we have only found three similar studies exploring the association between solid fuel and depression whose findings were consistent with our results that individuals using solid fuel were at a higher risk of depression. [17-19] Besides, regarding household air pollution, there is an increasing link between mental diseases and household air pollutants. This study contributes to the limited literature on the association between solid fuel and poor sleep quality in the elderly. Our results indicated that using solid fuel was significantly associated with poor sleep quality, even after accounting for a wide range of covariates, including socio-demographic factors, lifestyle behaviors, and presence of chronic diseases. Our findings were in line with previous findings. [20-22] However, limited evidence is available on the association between mental diseases and household air pollutants in pathogenesis research. One possible reason is that solid fuel combustion produces much higher levels of various gaseous pollutants than clean fuel and may increase the risk of developing mental disorders such as depression and poor sleep quality through pathways such as cerebrovascular damage, oxidative stress, neuroinflammation, or neurodegeneration. [5, 6] (P8, L228-247)

4.4. Cognition as the protective factor linking systolic blood pressure to self-rated health

An indirect path linking solid fuel and self-rated health was identified through cognitive abilities, although the other mediators including systolic blood pressure and BMI were not found to be associated with self-rated health. Our findings suggested that people who use solid fuel for cooking had lower cognitive abilities (b=-0.319, S.E.=0.111). Several studies in China contributed to the literature on the association between solid fuel and poor cognitive ability in the elderly. Recent evidence from a study in China found that solid fuel use was significantly associated with mental health and cognitive ability in middle-aged and older adults [2]. There were also studies that further showed the potentially harmful effects of household air pollution exposure on other aspect of memory. A follow-up study showed that solid fuel use was associated with a greater decline in cognitive score, and mostly in the episodic memory and visuo-construction dimensions[24]. Another prospective analysis found a significant adverse impact on cognitive abilities, especially short-term memory and mathematical reasoning. These results demonstrated that using solid fuel poses a health threat to elderly people [25]. (P8, L248-262)

4.5. Strengths and limitations

This study had several strengths. First, this study specified the pathways of the relationship between solid fuel use and self-rated health of the elderly. Second, self-rated health was not only considered a comprehensive measurement indicator of health, but it is also used as a predictor of morbidity and mortality [26]. This study’s results can help achieve primary and secondary prevention and improve the health of the elderly by stopping the use of solid fuel. Lastly, our study included 500 sample areas in 23 provinces in China, which gives our findings a strong external validity for the Chinese society. Moreover, the findings were robust across the study regions, demographic characteristics, and lifestyle behaviors.

However, a major drawback should be noted. Assessments for self-rated health in our study was based on only one self-reported question, and this may not provide an exactly accurate information compared with a structured interview tool. Other limitations should also be mentioned. First, constrained by the CLHLS data, we were unable to derive information on whether the individuals were responsible for cooking, types of cooking in childhood, and time spent cooking with indoor solid fuel; additionally, indoor air pollution exposure may vary by family and personal characteristics [23]. Second, the results of this cross-sectional study may not explain the underlying mechanisms of the relationship between solid fuel use and self-rated health. The underlying mechanisms may need to be investigated in large prospective cohort studies. Third, the relationship of mediators including sleep quality, cognitive ability, depression, systolic blood pressure, and BMI with solid fuel use has been confirmed in previous studies. Other diseases or pathophysiological indicators that may be caused by the use of solid fuel, which have not been proven, can be explored in future studies. (P8-9, L263-287)

REFERENCES

  1. Dutta, A.;Mukherjee, B.;  Das, D.;  Banerjee, A.; Ray, M. R., Hypertension with elevated levels of oxidized low-density lipoprotein and anticardiolipin antibody in the circulation of premenopausal Indian women chronically exposed to biomass smoke during cooking. Indoor Air 2011,21 (2), 165-76.
  2. Luo, Y.;Zhong, Y.;  Pang, L.;  Zhao, Y.;  Liang, R.; Zheng, X., The effects of indoor air pollution from solid fuel use on cognitive function among middle-aged and older population in China. Sci Total Environ 2021, 754, 142460.
  3. Yu, Q.; Zuo, G., Relationship of indoor solid fuel use for cooking with blood pressure and hypertension among the elderly in China. 2022.
  4. Sun, J.;Wang, J.;  Shen, Z.;  Huang, Y.;  Zhang, Y.;  Niu, X.;  Cao, J.;  Zhang, Q.;  Xu, H.;  Zhang, N.; Li, X., Volatile organic compounds from residential solid fuel burning in Guanzhong Plain, China: Source-related profiles and risks. Chemosphere 2019, 221, 184-192.
  5. Trishul Siddharthan, M. R. G., Dina Goodman,Muhammad Chowdhury, Association between Household Air Pollution Exposure and Chronic Obstructive Pulmonary Disease Outcomes in 13 Low- and Middle-Income Country Settings. American Journal of Respiratory & Critical Care Medicine 2018.
  6. Block, M. L.; Calderón-Garcidue?As, L., Air pollution: mechanisms of neuroinflammation and CNS disease. Trends in Neurosciences 2009, 32(9), 506-516.
  7. Ellen Idler, K. C., What Do We Rate When We Rate Our Health? Decomposing Age-related Contributions to Self-rated Health. Journal of Health and Social Behavior 2018, 59, 74-93.
  8. Engberg, I. S., J.Waller, G.Wennberg, P.Eliasson, Fatigue in the general population- associations to age, sex, socioeconomic status, physical activity, sitting time and self-rated health: the northern Sweden MONICA study 2014. BMC Public Health 2017, 17 (1), 654.
  9. Meyer, O. L.;Castro-Schilo, L.; Aguilar-Gaxiola, S., Determinants of mental health and self-rated health: a model of socioeconomic status, neighborhood safety, and physical activity. Am J Public Health 2014, 104 (9), 1734-41.
  10. Zeng Yi, J. W. V., Functional Capacity and Self-Evaluation of Health and Life of Oldest Old in China. Journal of Social Issues 2002, 58, 733-748.
  11. Xiaozhen;Wenyuan;  Yuan;  Huashuai;  Chen;  Zeng;  Xin;  Albert;  Hofman; Huali, Cognitive decline and mortality among community-dwelling Chinese older people. BMC medicine 2019.
  12. Zeng, Y.;Feng, Q.;  Hesketh, T.;  Christensen, K.; Vaupel, J. W., Survival, disabilities in activities of daily living, and physical and cognitive functioning among the oldest-old in China: a cohort study. The Lancet 2017, 389 (10079), 1619-1629.
  13. Elena M.Andresen, J. A. M., William B.Carter, Screening for Depression in Well Older Adults: Evaluation of a Short Form of the CES-D. American Journal of Preventive Medicine 1994, 10, 77-84.
  14. Cheng, S.; Chan, A., The Center for Epidemiologic Studies Depression Scale in older Chinese: thresholds for long and short forms. International Journal of Geriatric Psychiatry 2010, 20.
  15. Jylh, M., What is self-rated health and why does it predict mortality? Towards a unified conceptual model. Social Science & Medicine 2009, 69(3), 307-316.
  16. Idler, E. L.; Benyamini, Y., Self-rated health and mortality: a review of twenty-seven community studies. Journal of Health & Social Behavior 1997, 38 (1), 21-37.
  17. Shao, J.;Ge, T.;  Liu, Y.;  Zhao, Z.; Xia, Y., Longitudinal associations between household solid fuel use and depression in middle-aged and older Chinese population: A cohort study. Ecotoxicol Environ Saf 2021, 209, 111833.
  18. Li, C.;Zhou, Y.; Ding, L., Effects of long-term household air pollution exposure from solid fuel use on depression: Evidence from national longitudinal surveys from 2011 to 2018. Environ Pollut 2021, 283, 117350.
  19. Liu, Y.;Chen, X.; Yan, Z., Depression in the house: The effects of household air pollution from solid fuel use among the middle-aged and older population in China. Sci Total Environ 2020, 703, 134706.
  20. Haqing, Y., Solid Fuel-Related Indoor Air Pollution And Poor Sleep Quality In Adults Aged 45 Years And Older. Public Health Theses 2007.
  21. Yu, H.;Luo, J.;  Chen, K.;  Pollitt, K. J. G.; Liew, Z., Solid fuels use for cooking and sleep health in adults aged 45 years and older in China. Sci Rep 2021, 11 (1), 13304.
  22. Chen, C.;Liu, G. G.;  Sun, Y.;  Gu, D.;  Zhang, H.;  Yang, H.;  Lu, L.;  Zhao, Y.; Yao, Y., Association between household fuel use and sleep quality in the oldest-old: Evidence from a propensity-score matched case-control study in Hainan, China. Environ Res 2020, 191, 110229.
  23. Balakrishnan K, R. P., Sambandam S, Air pollution from household solid fuel combustion in India: an overview of exposure and health related information to inform health research priorities. Global Health Action 2011, 4, 1-9.
  24. Cao, L.;Zhao, Z.;  Ji, C.; Xia, Y., Association between solid fuel use and cognitive impairment: A cross-sectional and follow-up study in a middle-aged and older Chinese population. Environ Int 2021, 146, 106251.
  25. Qiu, Y.;Yang, F.-A.; Lai, W., The impact of indoor air pollution on health outcomes and cognitive abilities: empirical evidence from China. Population and Environment 2019, 40 (4), 388-410.
  26. Kaplan, G. A., Goldberg, D. E., Everson, Perceived Health Status and Morbidity and Mortality: Evidence from the Kuopio Ischaemic Heart Disease Risk Factor Study. international journal of epidemiology 1996, 25, 259-265.

Reviewer 3 Report

This paper conducts a study on the association between the use of solid fuel and health in people over 65 years of age in China. The structure of the article is good, and the methodology is regular. I consider that the article is not relevant for this journal. In addition, the findings may be modified by a multitude of variables that are not controlled. So I think this article should be rejected.

Author Response

  1. This paper conducts a study on the association between the use of solid fuel and health in people over 65 years of age in China. The structure of the article is good, and the methodology is regular. I consider that the article is not relevant for this journal. In addition, the findings may be modified by a multitude of variables that are not controlled. So I think this article should be rejected.

[Response]: Thank you for your valuable suggestion. According to your suggestion, key confounding factors mentioned in previous studies including socio-demographic factors (age, sex, marital status, community location, education years and household income) and lifestyle behaviors (smoking status, alcohol use and regular exercise) have been incorporated into the structural equation model [1-3], and this has been written in the covariates section. (P4, L118-120)

Socio-demographic factors: Studies have shown that solid fuels have a greater impact on increased blood pressure and cognitive decline in middle-aged and elderly people over 45 years old. Notably, in most Chinese households, women are the primary cooks, meaning that elderly women are exposed to air pollution for a longer time than men, giving them higher exposure levels. In China, more people use solid fuels for cooking in rural areas than in urban areas. Groups with higher education levels and higher annual income are usually more willing to use clean fuel.

Lifestyle behaviors: Alcohol use and regular exercise are commonly used covariates in studies of the effects of solid fuel use on physical health. A study has reported that smoking and solid fuel use had an interactive effect on BMI [4].

REFERENCES

  1. Ellen Idler, K. C., What Do We Rate When We Rate Our Health? Decomposing Age-related Contributions to Self-rated Health. Journal of Health and Social Behavior 2018, 59, 74-93.
  2. Engberg, I. S., J.Waller, G.Wennberg, P.Eliasson, Fatigue in the general population- associations to age, sex, socioeconomic status, physical activity, sitting time and self-rated health: the northern Sweden MONICA study 2014. BMC Public Health 2017, 17 (1), 654.
  3. Meyer, O. L.;Castro-Schilo, L.; Aguilar-Gaxiola, S., Determinants of mental health and self-rated health: a model of socioeconomic status, neighborhood safety, and physical activity. Am J Public Health 2014, 104 (9), 1734-41.
  4. Pan, M.;Gu, J.;  Li, R.;  Chen, H.;  Liu, X.;  Tu, R.;  Chen, R.;  Yu, S.;  Mao, Z.;  Huo, W.;  Hou, J.; Wang, C., Independent and combined associations of solid-fuel use and smoking with obesity among rural Chinese adults. Environ Sci Pollut Res Int 2021.

Round 2

Reviewer 2 Report

The revisions made by the authors are satisfactory in my opinion. I have no further major changes or corrections to suggest.

Reviewer 3 Report

The changes introduced by the authors are relevant and increase the validity of the study.